# Blue Phosphorescent Pt(II) Compound Based on Tetradentate Carbazole/2,3′-Bipyridine Ligand and Its Application in Organic Light-Emitting Diodes

**DOI:** 10.3390/molecules29122929

**Published:** 2024-06-20

**Authors:** Hakjo Kim, Chan-Hee Ryu, Miso Hong, Kang Mun Lee, Unhyeok Jo, Youngjin Kang

**Affiliations:** 1Division of Science Education, Department of Chemistry, Kangwon National University, Chuncheon 24341, Republic of Korea; 2School of Chemical Engineering, Sungkyunkwan University, 2066 Seobu-ro, Jangan-gu, Suwon 16419, Republic of Korea

**Keywords:** tetradentate ligand, photoluminescent, ligand-centered, quantum efficiency, PHOLEDs

## Abstract

The tetradentate ligand, merging a carbazole unit with high triplet energy and dimethoxy bipyridine, renowned for its exceptional quantum efficiency in coordination with metals like Pt, is expected to demonstrate remarkable luminescent properties. However, instances of tetradentate ligands such as bipyridine-based pyridylcarbazole derivatives remain exceptionally scarce in the current literature. In this study, we developed a tetradentate ligand based on carbazole and 2,3′-bipyridine and successfully complexed it with Pt(II) ions. This novel compound (**1**) serves as a sky-blue phosphorescent material for use in light-emitting diodes. Based on single-crystal X-ray analysis, compound **1** has a distorted square-planar geometry with a 5/6/6 backbone around the Pt(II) core. Bright sky-blue emissions were observed at 488 and 516 nm with photoluminescent quantum yields of 34% and a luminescent lifetime of 2.6 μs. TD-DFT calculations for **1** revealed that the electronic transition was mostly attributed to the ligand-centered (LC) charge transfer transition with a small contribution from the metal-to-ligand charge transfer transition (MLCT, ~14%). A phosphorescent organic light-emitting device was successfully fabricated using this material as a dopant, along with 3′-di(9H-carbazol-9-yl)-1,1′-biphenyl (mCBP) and 9-(3′-carbazol-9-yl-5-cyano-biphenyl-3-yl)-9H-carbazole-3-carbonitrile (CNmCBPCN) as mixed hosts. A maximum quantum efficiency of 5.2% and a current efficiency of 15.5 cd/A were obtained at a doping level of 5%.

## 1. Introduction

Recently, there has been significant interest in blue and sky-blue phosphorescent platinum complexes featuring tetradentate ligands. This interest is primarily because of their excellent performance in organic light-emitting devices (OLEDs), characterized by high external quantum efficiency (EQE) and long operational lifetimes [1,2,3,4,5,6,7,8,9,10,11,12,13,14,15]. The complexation of Pt(II) ions with tetradentate ligands results in the formation of five- or six-membered metallacycles. Platinum compounds with tetradentate ligands often exhibit unique structural characteristics. For example, three rings around the Pt(II) core comprising a combination of a five- and two six-membered rings, or two five-membered rings and a six-membered ring, or three six-membered rings, are continuously fused to each other, resulting in a highly rigid structure. In such cases, the primary coordination mode of the ligand is typically C^N or C^C chelation (where C represents a phenyl or *N*-heterocyclic carbene and N represents pyridine, pyrazole, etc.), as shown in [Fig molecules-29-02929-ch001] [16,17,18].

These compounds are widely applied in various informational devices, such as OLEDs, and in organic lighting. For instance, blue phosphorescent organic light-emitting diodes (PHOLEDs) based on a 3′-di(9H-carbazol-9-yl)-1,1′-biphenyl (mCBP) and 9-(3′-carbazol-9-yl-5-cyano-biphenyl-3-yl)-9H-carbazole-3-carbonitrile (CNmCBPCN) mixed host doped with a platinum(II) 9-(pyridin-2-yl)-2-(3-(3-methyl-1*H*-1,2,4-triazol-5-yl)phenoxy)-9H-carbazole analogue (left in [Fig molecules-29-02929-ch001]) exhibit a maximum external quantum efficiency (EQE) of 26.6% and a power efficiency (PE) of 64.3 lm W^–1^ [17]. Our group also reported Pt(II)-based triplet emitters ([1,1-bis(2′,6′-dialkyl-2,3′-bipyridyl-*N*,*C*^4^′)-1-methoxyethane]platinum(II), middle in [Fig molecules-29-02929-ch001]) with two C^N chelate 2,3′-bipyridine connected by a carbon linker. The blue emissive PHOLED incorporating (9-(3-(9H-carbazol-9-yl) phenyl)-9Hcarbazol-3-yl) diphenylphosphine oxide (mCPPO1) as a host in an EML realized good OLED performance, with a power efficiency of 24.9 lm W^–1^ and an EQE of 17.6% [18]. Moreover, Li et al. achieved an EQE of 24.4% and blue PHOLEDs with lifetimes of greater than 30,000 h (LT 70) at 100 cd m^−2^ using mCBP as a host and two carbazole-based Pt(II) compounds as dopants (platinum(II) 9-(pyridin-2-yl)-2-(9-(pyridin-2-yl)-9H-carbazol-2-yloxy)-9H-carbazole PtNON, right in [Fig molecules-29-02929-ch001]) [19]. This study demonstrated that the stability and role of platinum dopants containing tetradentate ligands are crucial for achieving a long lifespan.

The key electronic transitions of tetradentate-based phosphorescent Pt(II) complexes originate from a triplet ligand-centered transition (^3^LC) or inter ligand charge transfer (^3^ILCT) mixed with a metal-to-ligand charge transfer transition (^3^MLCT) [20,21,22,23,24]. Therefore, to achieve blue emission, a chelation ligand with large singlet and triplet energies is typically employed. Moreover, the ligand must possess appropriate coordination atoms capable of forming both five- and six-membered metallacycles. Therefore, carbazole-based ligands with large triplet energies, particularly pyridinyl-carbazole, have recently received considerable attention as tetradentate ligands [25,26,27]. Considering the aforementioned criteria for achieving blue emission, bipyridine derivatives with C^N chelation modes are highly recommended ligands. They exhibit high triplet energies and quantum efficiencies when metallated with heavy Ir(III) or Pt(II) ions [28,29,30,31,32,33,34,35,36]. However, examples of tetradentate ligand production through the combination of bipyridine and carbazole are extremely rare. To date, there have been no examples of a tetradentate ligand bearing both bipyridine and pyridinyl-carbazole groups. The tetradentate ligand, which combines a carbazole unit with high triplet energy and dimethoxy bipyridine, which is known for its high quantum efficiency when coordinated with a metal (Pt), is anticipated to exhibit promising luminescent properties. Therefore, we designed and developed a new tetradentate ligand, 2-((2′,6′-dimethoxy-[2,3′-bipyridin]-6-yl)oxy)-9-(pyridin-2-yl)-9H-carbazole (pypyOczpy). In this report, we describe the synthesis of the new ligand, its utilization in the synthesis of a platinum compound, the fabrication of an electroluminescent device using this compound, and an examination of its electroluminescent properties.

## 2. Results and Discussion

### 2.1. Synthesis and Crystal Structure

Compound **1** was obtained by the reaction of Pt(COD)Cl_2_ with tetradentate pypyOczpy in the presence of Na(OAC) in good yields (70%), as depicted in Figure 1. The structure of **1** was confirmed by various spectroscopic methods, including single-crystal X-ray analysis. Figure 1 illustrates the crystal structure of **1** in the asymmetric unit. The detailed parameters, including selected bond lengths and angles, are provided in Appendix A. The crystal structure of **1** clearly exhibits tetra-coordination around the Pt(II) center with a square-planar geometry. The bond lengths (ca. 2.0 Å, Appendix A) of both Pt-C and Pt-N were found to be very similar to those of previously reported square-planar Pt(II) compounds [37,38,39]. In addition, the pyridine group coordinated with the Pt(II) center was significantly distorted (Figure 1b, side view) because of severe steric hindrance from the methoxy-substituted pyridine moiety. The distortion was confirmed by the observation that the sum of the angles around the Pt(II) center (361.12°, Appendix A) was >360°. This suggests that the structural variation and vibrational motion centered on Pt(II) may be considerably restricted. Interestingly, the dihedral angle between two planes (bipyridine and pyridine ring of pyridyl-carbazole) is approximately 33.82° (See Appendix A). Therefore, distorted square-planar geometry can be due to high distortion between bipyridine and the pyridine-attached carbazole.

Furthermore, the molecular packing system for **1** in a simple unit cell demonstrated that the intermolecular distances were over 3.6 Å (Appendix A). This feature indicates that the intermolecular interactions of **1**, which could induce aggregation and excimer formation, were significantly inhibited in the solid state. The distance between Pt atoms in two adjacent molecules was approximately 7.8 Å, which was too far to suggest any significant interaction between the two Pt atoms (See Appendix A).

Thermogravimetric analysis (TGA) experiments were conducted to investigate the thermal stability of **1**. Owing to the presence of solvent molecules within the crystal lattice, a weight loss attributed to solvent molecules was observed up to approximately 250 °C. Taking this into account, a weight loss of approximately 5% was observed around 360 °C, defined as the typical decomposition temperature (See Appendix A). This result clearly supports the fact that this compound possesses sufficient thermal stability for the fabrication of OLEDs.

### 2.2. Time Dependent-Density Functional Theory (TD-DFT) Calculation

Density functional theory (DFT) calculations were performed to determine the frontier molecular orbitals of **1** [40,41,42]. The highest occupied molecule orbital (HOMO) is predominantly located on the π orbitals originating the *C*-coordinating dimethoxypyridine and pyridinyl-carbazole ring, supplemented by a *d*-orbital contribution (13.1%) from the Pt(II) ion. Conversely, the lowest unoccupied molecule orbital (LUMO) is mainly located on the π* orbitals of the *N*-coordinating pyridine rings, which originate from both carbazole and dimethoxy-substituted bipyridine (Figure 2). At the LUMO level, the *d*-orbital contribution from the Pt(II) ion is almost negligible (1.1%). These results suggest that the electron transitions in compound **1** are mainly attributed to interligand charge transfer (ILCT) and metal-to-ligand charge transfer transitions. Similar calculation results were also observed for Pt(II) compounds containing tetradentate ligands based on the carbazole unit [43]. The S_0_ → S_1_ and S_0_ → T_1_ transitions from the HOMO to LUMO were estimated to be approximately 413 and 460 nm, respectively. These values are similar to the experimental observations (mentioned in the following section). The calculated energy levels for the HOMO and LUMO were −5.10 and −1.48 eV, respectively. However, the energies determined through cyclic voltammetry (CV, Appendix A) experiments (−5.56 eV for HOMO and −2.57 eV for LUMO) revealed some discrepancies with the calculated results (Figure 2). These discrepancies can be attributed to the differences in the geometry optimization and environmental conditions between the solid-state calculations and experimental measurements conducted in solution. Detailed calculation results are presented in Appendix A.

### 2.3. Photophysical Properties

Figure 3 shows the absorption and emission spectra of compound **1** in CH_2_Cl_2_ solution measured at 298 K and 77 K. Compound **1** exhibited strong absorption bands in the range 230–270 nm and moderate absorption at approximately 366 nm, and both of these points showed significantly high absorbance coefficients. The shapes of the absorption peaks and absorption spectra closely resembled those reported for platinum compounds (Pt(ppzOczpy), PtON1, Pt(ppzOczpy-m), Pt(ppzOczpy-2m), and Pt(ppzOczpy-4m) [43]. However, this indicated a general shift toward longer absorption wavelengths. In particular, the edge of the absorption appeared at longer wavelengths compared to the previous compounds, suggesting a lower S_1_ energy for this compound. According to previous reports, the stronger absorption band can mostly be attributed to the interligand charge transfer (ILCT), whereas the band with moderate absorption intensity at longer wavelengths can be attributed to singlet metal-to-ligand charge transfer (^1^MLCT) transitions [24,44]. This compound exhibits sky-blue phosphorescence at room temperature with a maximum emission wavelength of 486/518 nm. The typical excimer peak observed with increasing concentrations of square-planar platinum compounds was not observed for this compound. When films with 5 and 50% concentrations were prepared and the emission spectra were measured, no significant changes were observed in the maximum emission wavelength or shape of the spectrum, despite some differences in emission intensity (See Appendix A).

Interestingly, the well-resolved emission peak of this compound was more than 10 nm longer than that of PtNON (λ_max_ = 508 nm), with the two carbazole units connected by oxygen [19]. However, the shoulder peak at shorter wavelengths, which was not observed for PtNON, indicated that most of the phosphorescent emission in this compound could be attributed to the ligand-centered (LC) transition.

Well-structured emission peaks are also observed at 77 K (Figure 4). The triplet energy (T_1_) was determined using the emission maximum at 77 K, and the corresponding triplet energy was found to be 2.56 eV. This triplet energy was lower than those of the carbazole-based Pt(II) analog, PtNON (E_T_ = 2.83 eV), and difluoro-substituted bipyridine-based Pt(II) analog, [1,2-bis(2′,6′-difluoro-2,3′-bipyridin-N,C^4^′) phenyl] Pt(II) (E_T_ = 2.7 eV) [17,24]. However, the T_1_ of compound **1** was comparable to those of Pt(II) complexes bearing similar tetradentate dimethoxy-bipyridine and phenyl indazole ligands [13,14,24].

The emission spectra at 298 K and 77 K were overall very similar in terms of maximum emission wavelength, but two notable differences were observed (Figure 3). Firstly, the emission spectrum at 77 K was better structured compared to that at 298 K, indicating that ligand-centered emission occurs more readily at lower temperatures. Secondly, among the two maximum emission peaks, the short-wavelength peak at 77 K exhibited higher intensity compared to the short-wavelength peak at 298 K. This is likely due to the rigidochromic effect, which occurs as a result of the enhanced rigidity of the structure at lower temperatures [45]. 

The absolute photoluminescence quantum efficiency (PLQY) of **1** was determined by fabricating a thin film doped with 1,3-bis(N-carbazolyl)benzene (mCP). The measured PLQY of **1** was 0.34, which was considerably lower than that of other platinum compounds (0.60–0.83) based on tetradentate carbazole ligands [13,19,43]. The phosphorescence decay time constant (τ) at ambient temperature is 2.60 μs in a 5% doped PMMA film (Appendix A). Interestingly, compound **1** has shorter τ than other analogs, such as Pt(ppzOczpy) (τ = 7.0 μs), PtON1 (τ = 3.8 μs), PtON7 (τ = 4.1 μs), PtNON (τ = 3.76 μs), and Pt(azole/pycz) (τ = 4.9 μs) [13,17,19,43,46]. The lifetime of the excited state is closely related to quantum efficiency [47,48]. A longer excited-state lifetime for a particular molecule indicates the existence of numerous vibrational modes in the excited state, which may increase the possibility of thermal non-radiative decay, consequently leading to a decrease in quantum efficiency. To understand the low PLQY despite the short lifetime of **1** in the excited state, the rate constants, radiative decay (k_r_), and non-radiative decay (k_nr_) (s^−1^) were evaluated. The estimated k_r_ and k_nr_ constants were 1.3 × 10^5^ and 2.5 × 10^5^, respectively. The k_r_ values were highly comparable, but the k_nr_ of the synthesized compound was considerably greater than that of the carbazole-based Pt(II) analogs (k_nr_ ≈ 10^4^), indicating that the lower PLQY may be attributed to the enhancement of non-radiative decay processes [49]. Therefore, integrating the dimethoxypyridine unit into one of the tetradentate carbazole-based ligand frameworks exhibits certain weaknesses in terms of quantum efficiency.

### 2.4. PHOLEDs Performances

To evaluate the electroluminescence (EL) properties of compound **1**, we fabricated phosphor-doped devices based on an exciplex-free co-host with dopant doping concentrations of 3–20 wt%. The detailed materials of the common layers and the device structure with an energy-level diagram are illustrated in Figure 4 and explained in the experimental section. The current (J), voltage (V), and luminescence (L) characteristics, assessed according to the doping concentration, are presented in Figure 5a. All devices showed a low turn-on voltage of 3.5 V. This was attributed to efficient charge injection via the cascade energy level structure of the devices. The values of driving voltage (V_d_) corresponding to 1000 cd m^−2^ were 5.41/5.44/5.71/5.79 V for the 3/5/10/20 wt% doped devices, respectively. The increasing trend of V_d_ as a function of dopant concentration was attributed to the charge-trapping effect of compound **1**. Holes can be strongly trapped by compound **1** because of the large HOMO level gap between the host and phosphor, and the current density is reduced with increasing doping concentration at the same voltage. Figure 5b shows the EQE-L properties of these devices. The maximum EQE (EQE_max_) values of the 3/5/10/20 wt% phosphor-doped devices were 4.6/5.2/4.3 and 3.6%, respectively. The measured EQE_max_ exhibited a relatively low quantum efficiency compared to those of other reported platinum complexes. It was deduced that the values of EQE_max_ were a result of the low quantum efficiency of compound **1** because the EQE is strongly related to the PLQY. The highest EQE_max_ of 5.2% was observed for the 5 wt% doped device because of the alleviated doping concentration quenching compared to the 10 and 20 wt% doped devices. Although the 3 wt% doped device had a lower doping concentration than the 5 wt% doped device, the reduced EQE_max_ value was attributed to the inefficient energy transfer from the host to compound **1**. The power efficiency (PE) and L-current efficiency (CE) are shown in Figure 5c. The maximum PE (PE_max_)/CE (CE_max_) values were 11.9 lm W^−1^/13.5 cd A^−1^ for 3 wt%, 13.4 lm W^−1^/15.5 cd A^−1^ for 5 wt%, 10.9 lm W^−1^/13.0 cd A^−1^ for 10 wt%, and 9.1 lm W^−1^/11.1 cd A^−1^ for 20 wt% emitter-doped devices. The order of PE_max_ and CE_max_ can be attributed to the quantum efficiency of each device. The EL spectra of the devices are shown in Figure 5d. The structured EL spectra are similar to the PL spectra, with wavelength peaks at approximately 489/519 nm. In the 410 nm wavelength region, the EL intensities gradually decreased with respect to the doping concentration of compound **1**, indicating that the energy transfer process was effectively enhanced owing to an increase in dopant molecules. All EL characteristics are summarized in Table 1. 

## 3. Experimental Section

### 3.1. General Information

Details regarding experimental conditions, X-ray analysis, and device fabrication/measurement are provided in the Appendix A. According to previous facts, 6-bromo-2′,6′-dimethoxy-2,3′-bipyridine [24] and 9-(pyridin-2-yl)-9H-carbazol-2-ol [50,51,52] have been synthesized.

### 3.2. Synthesis

Synthesis of 2-((2′,6′-dimethoxy-[2,3′-bipyridin]-6-yl)oxy)-9-(pyridin-2-yl)-9H-carbazole (pypy Oczpy): We added to a 100 mL schlenk flask 9-(pyridin-2-yl)-9H-carbazol-2-ol (1.0 g, 3.84 mmol), 6-bromo-2′,6′-dimethoxy-2,3′-bipyridine (1.36 g, 4.61 mmol), CuI (0.073 mg, 0.384 mmol), 2-picolinic acid (0.094 g, 0.758 mmol), and K_3_PO_4_ (1.63 g, 7.68 mmol). The flask was evacuated and backfilled with nitrogen, and DMSO (15 mL) was then added under an N_2_ atmosphere. The reaction mixture was stirred at 368–378 K, still under nitrogen conditions, for 3 d. After cooling to room temperature, the mixture was poured into water (100 mL) and extracted with ethyl acetate (50 mL × 3). The combined organic layer was dried with anhydrous Na_2_SO_4_ and concentrated under reduced pressure. Purification by column chromatography (dichloromethane/n-hexane 1:10 and then 1:3 *v*/*v*) afforded the desired product as a white solid (yield 1.3 g, 72%). Colorless crystals of X-ray quality were obtained by slow evaporation of a dichloromethane/n-hexane solution (1:1, *v*/*v*). ^1^H NMR (400 MHz, CDCl_3_): δ 8.66 (ddd, *J* = 5.6, 2.0, 0.8 Hz, 1H), 8.21 (d, *J* = 8.0 Hz, 1H), 8.08 (s, 1H), 8.01 (s, 1H), 7.86 (td, *J* = 7.6, 1.6 Hz, 1H), 7.82 (d, *J* = 8.4 Hz, 1H), 7.79 (d, *J* = 8.0 Hz, 1H), 7.72 (d, *J* = 2.0 Hz, 1H),7.67 (t, *J* = 8.0 Hz, 1H), 7.62 (dd, *J* = 8.0, 0.8 Hz, 1H), 7.42 (td, *J* = 7.6, 1.2 Hz, 1H), 7.32 (td, *J* = 7.6, 0.8 Hz, 1H), 7.27 (td, *J* = 5.0, 1.2 Hz, 1H), 7.15 (dd, J = 7.6, 1.6 Hz, 1H), 6.71 (d, *J* = 8.4 Hz, 1H), 6.30 (d, *J* = 8.8 Hz, 1H), 4.02 (s, 3H), 3.93 (s, 3H); ^13^C NMR (100 MHz, CDCl_3_): δ163.4, 163.0, 160.1, 153.2, 152.2, 151.7, 149.7, 142.4, 140.4, 140.0, 139.8, 138.7, 125.9, 124.2, 121.5, 121.3, 121.2, 120.9, 120.0, 119.1, 118.3, 115.1, 113.4, 111.2, 108.3, 104.4, 101.8, 53.8, 53.6; HRMS (EI): found *m*/*z* 474. 

Synthesis of **1**: A solution of 2-((2′,6′-dimethoxy-[2,3′-bipyridin]-6-yl)oxy)-9-(pyridin-2-yl)-9H-carbazole (0.23 g, 0.5 mmol), Pt(COD)Cl_2_ (0.21 g, 0.55 mmol), and NaOAc (0.12 g, 1.5 mmol) in PhCN (5 mL) was refluxed at 180 °C for 16 h. The volatiles were removed with a rotary evaporator, affording dark yellow residue. The crude mixture was purified by column chromatography (DCM/n-hexane = 1:2, *v*/*v*), and then, diethyl ether was added to induce the precipitate. By filtration, the Pt(pypyOczpy) was obtained as a yellow solid. Yield = 70%. ^1^H NMR (400 MHz, CD_2_Cl_2_) δ 9.64 (dd, *J* = 6.4, 1.2 Hz, 1H), 8.33 (d, *J* = 8.0 Hz, 1H), 8.18 (d, *J* = 8.0 Hz, 1H), 8.06 (d, *J* = 8.0 Hz, 1H), 7.96~8.00 (m, 2H), 7.91 (t, *J* = 8.0 Hz, 1H), 7.78 (d, *J* = 8.0 Hz, 1H), 7.45 (t, *J* = 8.0 Hz, 1H), 7.37 (t, *J* = 8.0 Hz, 2H), 7.09 (d, *J* = 8.0 Hz, 1H), 7.03 (td, *J* = 6.8, 1.2 Hz, 1H), 6.64 (s, 1H), 4.10 (s, 3H), 3.94 (s, 3H) ^13^C NMR (100 MHz, CD_2_Cl_2_) δ 189.5, 166.0, 162.7, 159.4, 157.8, 156.7, 151.3, 148.6, 141.3, 139.4, 138.9, 138.7, 128.6, 124.8, 123.0, 120.9, 120.5, 119.5, 119.4, 116.5, 116.0, 115.9, 115.7, 114.5, 111.2, 110.3, 106.2, 53.2, 53.1. Anal. calcd for C_29_H20N_4_O_3_Pt; C, 52.18; H, 3.02; N, 8.39; found: C, 52.20, H 3.05, N 8.36%.

### 3.3. Device Fabrication and EL Measurements

The device structure is composed of ITO (50 nm)/PEDOT: PSS (40 nm)/TAPC (10 nm)/oCBP (10 nm)/mCBP: CNmCBPCN: compound **1** (25 nm: 50 wt%:x wt%)/TSPO1 (5 nm)/TPBi (20 nm)/LiF (1.5 nm)/Al (200 nm), where ITO is indium tin oxide, PEDOT: PSS is poly(3,4-ethlyenedioxythiophene): poly(styrenesulfonate), TAPC is 4,4′-cyclohexylidenebis[N,N-bis(4-methyl-phenyl)aniline], oCBP is 2,2′-di(9H-carbazol-9-yl)biphenyl, mCBP is 3,3-di(9H-carbazol-9yl)biphenyl, CNmCBPCN is 9-(3′-(9H-carbazol-9-yl)-5-cyano-[1,1′-biphenyl]-3-yl)-9H-carbazole-3-carbonitrile, TSPO1 is diphenyl[4-(triphenylsilyl)phenyl]phosphine oxide, and TPBi is 2,2′,2″-(1,3,5-benzinetriyl)-tris(1-phenyl-1-H-benzimidazole), LiF is lithium fluoride, and Al is aluminum. All devices were fabricated by vacuum thermal evaporation under a high pressure of 5 × 10^−7^ after pre-treatment under ultraviolet ozone of ITO and spin-coating of PEDOT: PSS. Keithley 2400 source meter and CS2000 spectroradiometer were used to measure the electrical and optical EL performance, respectively.

## 4. Conclusions

We designed a novel tetradentate ligand that incorporates carbazole and dimethoxy-substituted bipyridine, enabling two C^N chelation modes for metal ions. Using this ligand, a new sky-blue phosphorescent Pt complex was synthesized. This compound has a distorted square-planar geometry around the central Pt ion with bite angles ranging from 81.43(14)° to 98.85(14)°. This compound exhibits sky-blue phosphorescence at room temperature, with a maximum emission wavelength of 486/518 nm and a quantum efficiency of 0.34. Despite increasing the concentration of this compound, the typical excimer formation commonly observed in Pt(II) compounds was hardly detected. From TD-DFT calculations, it was concluded the electronic transition for this compound might arise from ILCT (π_bpy/cz_ − π*_cz-py_) mixed with a small contribution of MLCT (Pt_d_ − π*_bpy/cz-py_). PHOLEDs were successfully fabricated using a mCBP: CNmCBPCN mixed host and this compound as a dopant. PHOLEDs with 5 wt% exhibited sky-blue emission with a maximum EQE of 5.2% and CIE_xy_ values of 0.24 and 0.53. The combination of dimethoxy-substituted bipyridine and carbazole offers advantages, such as the inhibition of excimer formation; however, it may pose some disadvantages in terms of low quantum efficiency. Overall, this study highlights the significance of designing chelating ligands capable of efficiently incorporating carbazole combinations into tetradentate ligands, further forming complexes with Pt. To enhance PHOLEDs’ performance, ongoing studies are focused on the synthesis of new Pt(II) compounds, including various bipyridine derivatives.

## Data Availability

Data generated or analyzed during this study are provided in full within the published article and its Appendix A.

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
