# Peer review of "Blue Phosphorescent Pt(II) Compound Based on Tetradentate Carbazole/2,3′-Bipyridine Ligand and Its Application in Organic Light-Emitting Diodes"

_molecules, 2024, doi:10.3390/molecules29122929_

Round 1

Reviewer 1 Report

Comments and Suggestions for Authors

This is a nice piece of work. A minor revision is recommended with some revision suggestions given below.

In the abstract, this sentence can be reworded for clarity: instances of tetradentate ligands such as bipyridine remain exceptionally scarce. Bipyridine is a bidentate ligand. The same problem for the sentence of “However, examples of tetradentate ligands such as bipyridine are extremely rare” in Introduction.

The three complexes in Chart 1 should be discussed in more details, including structural features and emission properties, and device performance. A proper compound name/abbreviation can be given for each compound. I am not sure if the compound referred to as PtNON in page 5 is one of the compounds shown in Chart 1.

The emission of Figure 4 can be combined with Figure 3 for a clear comparison of the emission spectra between rt and 77 K. Some discussion is needed for the comparison.

Please combine the subfigures of Figurer 6 into a complete figure.

The following related work is suggested being cited: Organometallics 2021, 40(2), 156-165.

Reviewer 2 Report

Comments and Suggestions for Authors

1. The title is not any attractive, pls revise it.

2. Why is novel? How could you control this novel?

3. The preparation and formation of the full ms is badly.

4. Pls do and discussion of the packing model of this complex.

5. Density functional theory (DFT) calculations were performed, some refs may be considered, such as Monatsh. Chem, 2019, 150, 1355–1364 and Theor. Chem. Acc. 2022, 141, 68.

6. The figure 5 is not right, pls check it.

7. The conclusion is complicated, pls improved it.

Comments on the Quality of English Language

revise 

Reviewer 3 Report

Comments and Suggestions for Authors

The authors of a manuscript as the tittle “Development of Carbazole and 2,3′-Bipyridine Based Tetradentate Ligand and Its Complexation with Pt(II) Ion for Creating Blue Phosphorescent Material” describe a continuation a novel tetradentate ligand that incorporates carbazole and dimethoxy-substituted bipyridine, enabling two C^N chelation modes for metal ions. Using this ligand, a new sky-blue phosphorescent Pt complex was synthesized. Professor Kang and co-workers carried out a comprehensive study of the tetradentate Pt complex by X-ray, cyclic voltammetry, photophysical properties and theoretical calculation studies. In addition, successfully fabricated a phosphorescent organic light-emitting device using this material as a dopant, together with 3'-di(9H-carbazole-9-yl)-1,1'-biphenyl (mCBP) and 9-(3'-carbazole-9-yl-5-cyano-biphenyl-3-yl)-9H-carbazole-3-carbonitrile (CNmCBPCN) as a mixed host. I find this study perfectly suitable for publication in Molecules with modifications.

Before publication, the authors should address the following minor technical points:

1. p1, line 17: Can you further explain the distortion of compound 1 of the square planar geometry, which is the 5/6/6 around the Pt(II) core.

2. Full description of the synthesis of the new ligand 2-((2',6'-dimethoxy-[2,3'-bipyridin]-6-yl)oxy)-9-(pyridin-2-yl)- 9H-carbazole (pypyOczpy) (detailed experimental, including yields and corresponding ESI spectra).

3. p2, line 65: Why do you need such a high temperature to obtain the Pt complex? Have you tried with other solvents than benzonitrile since it is not a very usual solvent in synthesis. If you did it in dmso, it would be obtained.

4. p2, line 78: In the X-ray structure, how much is the Pt-Pt distance between the Pt monomers. Some stacking is observed since in the emission spectra dimer formation would be observed.

5. p5, line 170: What concentration of compound 1 has been measured and at what wavelength you have irradiated, indicate it in Figure 3.

6. p6, line 171: What concentration of compound 1 has been measured and at what wavelength you have irradiated, indicate it in Figure 4.

7. p8, line 224: n-hexane

Comments on the Quality of English Language

The article is correctly written

Round 2

Reviewer 2 Report

Comments and Suggestions for Authors

accept

Author Response

I deeply appreciate your positive decision regarding our paper.